# UnMaskFork: Test-Time Scaling for Masked Diffusion via Deterministic Action Branching

**Kou Misaki** [1]   **Takuya Akiba** [1]

## Abstract

Test-time scaling strategies have effectively leveraged inference-time compute to enhance the reasoning abilities of Autoregressive Large Language Models. In this work, we demonstrate that Masked Diffusion Language Models (MDLMs) are inherently amenable to advanced search strategies, owing to their iterative and non-autoregressive generation process. To leverage this, we propose **UnMaskFork** (**UMF**), a framework that formulates the unmasking trajectory as a search tree and employs Monte Carlo Tree Search to optimize the generation path. In contrast to standard scaling methods relying on stochastic sampling, UMF explores the search space through deterministic partial unmasking actions performed by multiple MDLMs. Our empirical evaluation demonstrates that UMF consistently outperforms existing test-time scaling baselines on complex coding benchmarks, while also exhibiting strong scalability on mathematical reasoning tasks.

## 1. Introduction

The scaling laws of Large Language Models (LLMs) have recently expanded beyond pre-training parameters to inference-time compute. Specifically, test-time scaling (TTS) (Wang et al., 2023; Brown et al., 2024; Snell et al., 2025) demonstrates that allocating additional compute budget during inference, typically via Best-of-N or tree search, can significantly enhance reasoning capabilities in Autoregressive Models (AR-LLMs).

Parallel to this, Masked Diffusion Language Models (MDLMs) (Austin et al., 2021; Hoogeboom et al., 2021; Shi et al., 2024; Lou et al., 2024; Sahoo et al., 2024), which

model generation as an iterative transition from "mask" token sequence to clean text, have emerged as a compelling non-autoregressive alternative to AR-LLMs. Through large-scale pretraining, modern MDLMs (Nie et al., 2025; Ye et al., 2025; Gong et al., 2025; Xie et al., 2025) are approaching the performance of AR-LLMs of similar model size. However, compared to TTS research in AR-LLMs, unlocking the full reasoning potential of MDLMs through test-time scaling remains relatively unexplored.

In this work, we empirically demonstrate that applying standard AR-based scaling strategies, such as Best-of-N, where temperature is often increased to encourage diversity, is ineffective for MDLMs. Our extensive experiments on coding benchmarks reveal that increasing the temperature across the entire unmasking schedule degrades generation quality, thereby hindering performance improvements despite the increased diversity. We hypothesize that while stochastic sampling benefits autoregressive models, MDLMs rely on global iterative refinement, where early stochastic errors can be detrimental. In this context, early stochastic errors introduced by high-temperature sampling can propagate through subsequent denoising steps, disrupting global consistency and leading to irreversible structural defects. These observations underscore the need for a scaling paradigm that derives diversity from structural variations rather than stochastic noise injection.

To address this need, we propose **UnMaskFork** (**UMF**), a sample-efficient test-time scaling framework tailored for masked diffusion. Instead of relying on stochastic sampling within a single model, UMF achieves exploration through deterministic action branching. We formulate the unmasking trajectory as a search tree where branches represent distinct unmasking decisions made by different pre-trained MDLMs or deterministic strategies (e.g., varying inference parameters), and empirically verify that using multiple MDLMs leads to improvements. This approach replaces stochastic noise with deterministic actions that yield high-quality, distinct trajectories. Furthermore, this determinism allows for efficient node caching: UMF caches and reuses partially unmasked state, further optimizing the compute budget (defined by Number of Function Evaluations) during Monte Carlo Tree Search (MCTS) and consistently

[1]Sakana AI, Tokyo, Japan. Correspondence to: Kou Misaki <kou.misaki@sakana.ai>.

*Proceedings of the 43rd International Conference on Machine Learning*, Seoul, South Korea. PMLR 306, 2026. Copyright 2026 by the author(s).

outperforming other TTS baselines.

**Contributions.** ① We empirically show that temperature-based stochastic scaling in MDLMs is often less sample-efficient for the budget regimes we have tested, and that aggressive stochasticity can degrade generation quality. ② We propose **UMF**, an inference method that scales masked diffusion inference by exploring a tree of unmasking trajectories. By treating distinct MDLMs or inference parameters as discrete actions in MCTS, we achieve diverse exploration without sacrificing generation quality. ③ We demonstrate that UMF consistently outperforms existing baselines, such as Best-of-N and DTS*, on complex coding benchmarks such as LiveCodeBench, HumanEval+, and MBPP+, and we also show that UMF is effective in mathematical reasoning tasks.

## 2. Related Work

**Inference-time scaling and alignment for diffusion models.** For diffusion models, inference-time scaling has been explored through sampler modifications and longer/structured sampling procedures. In discrete masked diffusion, ReMDM (Wang et al., 2025) introduces principled remasking to allow tokens to be revisited and corrected, enabling improved quality as the number of sampling steps increases. Another line of work treats inference-time alignment to reward functions as a sampling/search problem. Similarly, PG-DLM (Dang et al., 2026) applies particle Gibbs and conditional SMC kernels to resample entire denoising trajectories, enabling trajectory-level refinement under reward guidance without retraining. Finally, the broader "diffusion + MCTS" trend also appears in planning, where diffusion-based trajectory generators are combined with MCTS-style search for improved test-time planning scalability (Yoon et al., 2025).

**Tree search for diffusion language model inference.** Recent work has started to apply explicit tree search to diffusion LM decoding itself, motivated by the combinatorial nature of choosing unmasking positions and committing tokens. Diffusion Tree Sampling (DTS) (Jain et al., 2025b) constructs a tree over the unmasking process and propagates terminal rewards to reuse past computation for scalable inference-time alignment. MEDAL (Huang et al., 2025) uses MCTS at the initialization stage to explore high-confidence unmasking trajectories and provide a stronger starting point for subsequent refinement. TReASURe (Yu et al., 2025) proposes a test-time alignment method tailored to masked diffusion, introducing branching and low-variance scoring mechanisms to address correlated branches and high-variance reward estimates when applying tree search to parallel unmasking. Our work complements these efforts by exploiting a distinct axis of exploration: Instead of relying primarily on stochastic branching within a single model, we define search actions at the level of selecting among multiple pretrained MDLMs (and deterministic inference configurations), enabling diverse yet high-quality partial unmasking decisions and efficient reuse of deterministic rollouts via caching. Structurally, we note that while DTS employs stochastic rollouts and incorporates all intermediate nodes along the path into the search tree, UMF utilizes deterministic rollouts that are cached for efficient reuse rather than being explicitly added to the search tree.

## 3. Preliminaries

In this section, we define the necessary notation and MDLM unmasking process to formulate the tree search in UMF.

### 3.1. Partially Masked State and Mask Ratio

Let $V$ be the vocabulary (i.e., the set of tokens), $m$ be the mask token, and $U := V \cup \{m\}$ be the extended vocabulary. Conditioned on a prompt token sequence $x^{\mathrm{prompt}} \in V^{n_{\mathrm{p}}}$ of length $n_{\mathrm{p}}$, the model generates $n_{\mathrm{g}}$ tokens. Let the total length be $n := n_{\mathrm{p}} + n_{\mathrm{g}}$, and denote the "partially-masked state" during inference as $z \in U^n$. The prompt is fixed such that $z_{0:n_{\mathrm{p}}-1} = x^{\mathrm{prompt}}$.

We define the set of mask positions in the generation segment (index set $\mathcal{I}_{\mathrm{g}} := \{n_{\mathrm{p}}, \ldots, n-1\}$) as $\mathcal{M}(z) := \{i \in \mathcal{I}_{\mathrm{g}} \mid z_i = m\}$. We also define the "residual mask ratio" of the generation segment as $\rho(z) := |\mathcal{M}(z)|/n_{\mathrm{g}}$. The initial state is $z_T = (x^{\mathrm{prompt}}, m, \ldots, m)$ (where the generation segment is fully masked), and the terminal state $z_0$ satisfies $\rho(z_0) = 0$ (i.e., fully unmasked).

### 3.2. MDLM Prediction and Unmask Transition

The MDLM predicts, for each position $i$, a categorical distribution $p_{\theta,i}(\cdot \mid z)$ over tokens conditioned on the current state $z$. Specifically, the model produces logits $\ell_{\theta,i}(\cdot \mid z)$, which are converted into a tempered distribution by scaling and normalizing: $p_{\theta,T,i}(x \mid z) := \mathrm{softmax}\left(\ell_{\theta,i}(x \mid z)/T\right)$. Here, $T = 1$ recovers the original model distribution, $T < 1$ sharpens it, and $T > 1$ flattens it. In the limit $T \to 0$, the distribution $p_{\theta,T,i}$ concentrates its probability mass on the token with the highest logit, corresponding to greedy selection ($x_i := \arg\max_x \ell_{\theta,i}(x \mid z)$). Conversely, for $T > 0$, we select tokens via stochastic sampling: $x_i \sim p_{\theta,T,i}(\cdot \mid z)$.

In a single unmasking step, given the current state $z_t$, we (1) obtain candidate tokens via model prediction and (2) select a subset of positions $S_t \subseteq \mathcal{M}(z_t)$ to commit (unmask and fix). Specifically, for $i \in S_t$, we sample $\hat{x}_i \sim p_{\theta,T,i}(\cdot \mid z_t)$ (or take the argmax) to construct the next state $z_{t-1}$: $z_{t-1,i} = \mathbb{I}(i \in S_t)\hat{x}_i + \mathbb{I}(i \notin S_t)z_{t,i}$. Positions not in $S_t$ remain unchanged. We assume *monotonic unmasking*, meaning positions are never re-masked once committed:

$\mathcal{M}(z_{t-1}) \subseteq \mathcal{M}(z_t)$.

### 3.3. Action as an Inference Configuration

In UMF, we formulate the search space not as a probabilistic branching over tokens, but as a discrete selection of inference configurations. We define an action $a \in \mathcal{A}$ as a tuple: $a := (\theta_a, T_a, g_a)$. Here, $\theta_a$ specifies the model parameters (i.e., selecting one of multiple pre-trained MDLMs), $T_a$ is the sampling temperature, and $g_a$ is the remasking strategy [1] (e.g., entropy-based or low-confidence) that determines the commit set $S_t$ from $z_t$. Given a state $z_t$, an action $a$ induces a transition $F_a$: $F_a : z_t \mapsto z_{t-1}$. Crucially, when using a low temperature $T_a \approx 0$ (greedy decoding) combined with a deterministic strategy $g_a$, the transition $F_a$ becomes fully deterministic, assuming fixed tie-breaking rules. This determinism is key to UMF, as it enables efficient node caching by avoiding redundant computations for identical state-action pairs.

### 3.4. Inference Budget (NFE)

We measure the computational cost using the Number of Function Evaluations (NFE). Formally, NFE counts the total number of MDLM forward passes required to compute the distribution $p_{\theta_a, i}(\cdot \mid z)$. A single unmasking step typically consumes 1 NFE. However, a key advantage of UMF is that if the transition or rollout result for a pair $(z, a)$ is retrieved from the cache, the NFE cost is zero.

### 3.5. Remasking Strategies ($g_a$)

The remasking strategy $g_a$ determines the subset of positions to be re-masked (or kept masked) based on the current state. Existing literature proposes both deterministic and stochastic approaches. Deterministic strategies typically target tokens with the lowest model confidence. Examples include the **Entropy-based strategy** (for Dream models (Ye et al., 2025)), which masks positions with the highest predictive entropy, and the **Low-confidence strategy** (for LLaDA models (Nie et al., 2025)), which masks positions where the model assigns the lowest probability to the selected token. Conversely, stochastic strategies, such as independent sampling ("origin") in Dream or random masking in LLaDA, introduce randomness into the selection process. We provide further details on these strategies in Appendix B.

## 4. Methods

### 4.1. Motivation

In this section, we propose **UnMaskFork** (**UMF**). Our method reformulates the inference process as a tree search

---

**Algorithm 1** UnMaskFork

```
 1: function UNMASKFORK(n_budget)
 2:     T, C ← INITTREE( ), INITCACHE( )
 3:     while NFE < n_budget do
 4:         N ← SELECT(T)
 5:         N_new, r ← EXPAND(N, T, C)
 6:         BACKUP(N_new, T, r)
 7:     return SELECTBEST(T)

 8: function EXPAND(N, T, C)
 9:     a ← SELECTROLLOUTACTION(N, T)
10:     if ISROLLOUTCACHED(C, N, a) then
11:         return GETCACHE(C, N, a)

12:     N_new ← UNMASKTONEXTRATIO(N, a, T)
13:     N_τ ← N_new
14:     SETNODECACHE(C, N_τ, a)
15:     while not FULLYUNMASKED(N_τ) do
16:         N_τ ← UNMASKTONEXTRATIO(N_τ, a, T)
17:         SETNODECACHE(C, N_τ, a)

18:     r ← EVALUATE(N_τ)
19:     SETSCORECACHE(C, N_τ, a, r)
20:     return N_new, r
```

---

where specific partial unmasking configurations serve as discrete actions within MCTS. UMF treats a high-performing inference configuration (e.g., a specific MDLM) as an atomic action. While random remasking induces diversity, it often degrades performance compared to adaptive strategies (such as low-confidence masking) that utilize inference-time information (Nie et al., 2025; Zhu et al., 2025; Kim et al., 2025). Therefore, we adopt confidence-based deterministic remasking strategies with temperature $T \approx 0$ [2] as our primary actions. This approach offers two major advantages: (1) it avoids performance degradation caused by suboptimal stochastic perturbations, as we empirically demonstrate in Section 6; and (2) it enables efficient node caching by leveraging the deterministic nature of low-temperature rollouts, sharply distinguishing UMF from stochastic baselines.

### 4.2. UnMaskFork

UMF formulates the unmasking trajectory as a search tree where nodes represent partially masked states at specific masking ratios, and branches correspond to selected inference actions. We employ MCTS to optimize the generation path. To strictly adhere to a fixed compute budget during Test-Time Scaling (TTS), we follow existing methodologies (Jain et al., 2025b) and measure the Number of Function Evaluations (NFE), terminating the search once the budget is exhausted. NFE corresponds to the number of forward passes of the model $p_\theta$. We expand nodes according to a discrete schedule of mask ratios $\rho$. In this work, we adopt the schedule $[0.9, 0.8, 0.7, 0.6, 0.5, 0.4, 0.2]$, which samples

---

[1] We use the term "remasking strategy" by convention, but we emphasize that in our case, unmasked tokens are never re-masked.

[2] For Dream and Dream-Coder, the recommended temperature $T = 0.1$ is used.

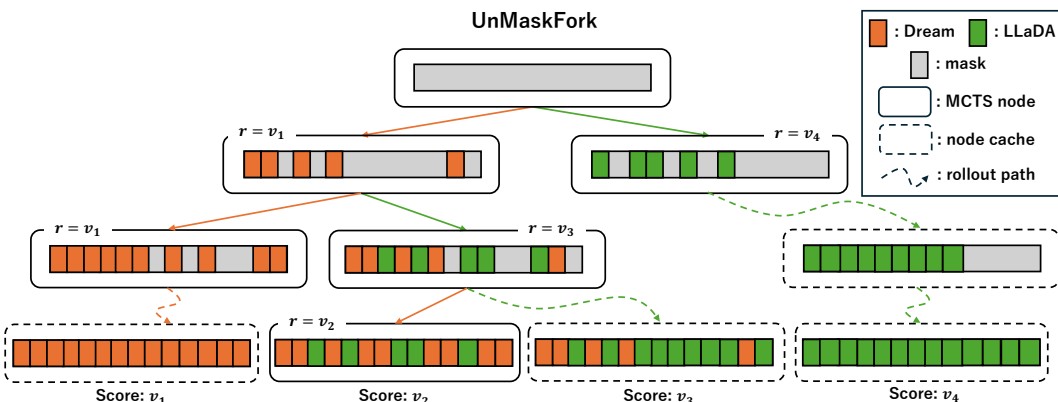

*Figure 1.* Conceptual diagram of UnMaskFork. Nodes generated during rollouts (dotted lines) and their evaluation results are cached to be reused in subsequent expansion steps, minimizing redundant computations.

more frequently in the early stages to ensure the diversity of the trajectory.

Algorithm 1 outlines the UMF procedure. One iteration of MCTS consists of three steps: *Select*, *Expand*, and *Backup*.

**Select.** In the *Select* step, we traverse the tree from the root to a node where unexplored actions remain. UMF selects a child node using the standard UCT score (Kocsis & Szepesvári, 2006): $\text{UCT}(N) = \frac{\sum_i r_i}{n_{\text{tgt}}} + c_{\exp}\sqrt{\frac{\log n_{\text{par}}}{n_{\text{tgt}}}}$, where $r_i$ represents rewards accumulated by backups, $n_{\text{par}}$ and $n_{\text{tgt}}$ are the visit counts of the parent node and the target node, respectively, and $c_{\exp}$ is the exploration coefficient. We set $c_{\exp} = 1$ in our experiments.

**Expand.** In the *Expand* step, we select an unexplored action $a$ from the chosen node $N$. In UMF, expanding a node entails advancing the state to the next predetermined residual mask ratio $\rho$. We achieve this via a procedure `UnMaskToNextRatio`, which iteratively applies atomic MDLM transitions (each consuming 1 NFE) while holding the action $a$ fixed until the target ratio is reached. Upon generating the new child node $N_{\text{new}}$, we immediately continue with a deterministic rollout until the sequence is fully unmasked to obtain a reward $r$. Crucially, throughout this process, we cache all intermediate nodes and the final reward $r$. Consequently, if a subsequent expansion visits a state-action pair $(N, a)$ that is already cached, we retrieve the trajectory and reward with zero effective NFE cost.

**Backup.** Once the reward $r$ is obtained, it is backed up to $N_{\text{new}}$ and all its ancestors. This value updates the node statistics used in the UCT score defined above for subsequent iterations.

### 4.3. Design of the Action Set

The design of the action set $\mathcal{A}$ is critical for performance. Extensive experiments in Section 6 confirmed that solely increasing temperature or randomizing mask order often hurts

performance in MDLMs. Therefore, we always include the highest-performing deterministic configuration (low temperature with greedy remasking) as one of the actions. Since the root node's action is selected early in the MCTS process, this design ensures that the strong unmasking trajectory is explored first, guaranteeing performance in low-budget regimes. Regarding other actions, while we demonstrate in Section 6.3.2 that switching between different pre-trained MDLMs yields the most significant gains, actions with different inference parameters (e.g., temperatures) can also improve performance over other TTS baselines.

**Handling heterogeneous tokenizers.** When using multiple MDLMs with different tokenizers (e.g., Dream (Ye et al., 2025) and LLaDA (Nie et al., 2025)), direct token transfer is infeasible. To address this, we explicitly map special tokens (MASK, EoS, Pad) directly between models to ensure control compatibility. For the non-special tokens, we employ a text-based mapping strategy: contiguous already-generated non-special segments are decoded into text via the source tokenizer and re-encoded by the target tokenizer. Across all experiments, the decoding budget remains fixed to a 768-token generation window. Tokenizer switching does not change this budget or the NFE schedule; only the tokenization length of already-generated non-special text spans can slightly drift, because the same string may be segmented differently by different tokenizers. This drift was small: on 100 LiveCodeBench problems at NFE=3072, the average relative length change was 1.09%. Empirically, we observed no structural degradation in the generated code resulting from this conversion. Furthermore, we emphasize that model switching does not occur at every denoising step, but only at the expansion nodes of the search tree. Given our maximum search depth of 7, the tokenizer is swapped at most 6 times throughout the entire 768-step generation process. Consequently, this infrequent switching introduces negligible instability compared to the benefit of utilizing diverse model priors.

# 5. Motivation and Analysis

In this section, we analyze the design choices of UMF. Specifically, we address two key design choices: (1) Why does forking the unmasking trajectory with different actions improve performance? (2) Why does UMF prioritize deterministic actions (e.g., switching models or heuristics) over standard stochastic sampling (e.g., increasing temperature), despite both offering diversity? We formulate these choices through the lens of diffusion kernel selection and sample efficiency under a fixed compute budget.

## 5.1. Inference as Adaptive Kernel Selection

The generation process of an MDLM can be viewed as a sequence of transitions using a reverse kernel $p_\theta(z_s|z_t)$. Following the formulation by Sahoo et al. (2024), the negative Evidence Lower Bound (ELBO) for the trajectory decomposes into a sum of KL divergences between the true posterior $q(z_s|z_t, x)$ and the model kernel $p_\theta$.

In the context of UMF, we define an action $a \in \mathcal{A}$ as a specific inference configuration, such as the choice of model parameters $\theta_a$ (e.g., distinct pre-trained models) or hyperparameters (e.g., temperature, remasking strategy). Selecting an action $a$ at step $t$ corresponds to choosing a specific kernel $K_t^a(\cdot|z_t)$ from a family of available kernels.

We can view the tree search as optimizing a state-dependent switching policy $\pi_t(a|z_t)$. Let $\epsilon_t^a(z_t)$ be the expected KL divergence error for action $a$ at state $z_t$. A switching policy that dynamically selects the best action can strictly outperform any single static model. Formally, relying on the inequality between the sum of minimums and the minimum of sums, we have:

$$\sum_t \mathbb{E}_{z_t}[\min_{a\in\mathcal{A}} \epsilon_t^a(z_t)] \leq \min_{a\in\mathcal{A}} \sum_t \mathbb{E}_{z_t}[\epsilon_t^a(z_t)]. \quad (1)$$

Eq. (1) implies that even if no single model is superior across all steps, a trajectory formed by interleaving the "best local kernels" achieves a lower accumulated error. This provides the theoretical motivation for exploring diverse actions. Importantly, this logic holds for any set of diverse kernels, whether they arise from stochastic perturbations or structural differences (e.g., multi-model).

## 5.2. Budget Efficiency: Deterministic vs. Stochastic Diversity

While both stochastic sampling (high temperature) and deterministic switching (multi-model) provide distinct unmasking trajectories, they differ fundamentally in their sample efficiency during tree search. This distinction is critical when operating under a fixed budget of Number of Function Evaluations (NFE).

In MCTS, we estimate the value $Q(z, a) = \mathbb{E}[R(\tau)|z, a]$ of

*Table 1.* Multi-task Pass@1 (%) for Best-of-N (BoN), DTS*, AB-MCTS, and UMF at NFE=12288.

| Method | LiveCodeBench | HumanEval+ | MBPP+ |
|---|---|---|---|
| BoN Pair (DCoder+LLaDA) | 19.0 | 75.0 | 66.0 |
| BoN (DCoder, entropy) | 18.0 | 71.0 | 66.0 |
| BoN (DCoder, origin) | 12.0 | 66.0 | 63.0 |
| BoN (LLaDA, low-confidence) | 13.0 | 63.0 | 45.0 |
| BoN (LLaDA, random) | 2.0 | 23.0 | 29.0 |
| DTS* Pair (DCoder+LLaDA) | 18.0 | 75.0 | 68.0 |
| DTS* (DCoder, entropy) | 18.0 | 72.0 | 67.0 |
| DTS* (DCoder, origin) | 10.0 | 65.0 | 60.0 |
| DTS* (LLaDA, low-confidence) | 14.0 | 63.0 | 47.0 |
| DTS* (LLaDA, random) | 1.0 | 27.0 | 31.0 |
| AB-MCTS (DCoder+LLaDA) | 21.0 | 81.0 | 68.0 |
| **UMF** | **28.0** | **88.0** | **72.0** |

a node by rolling out trajectories $\tau$ and observing the terminal reward $R$. The reliability of this estimation is governed by the variance of the rollout. Introducing stochasticity (e.g., $T > 0$) to generate diverse actions makes $R$ a random variable with variance $\text{Var}[R] > 0$. From standard Monte Carlo convergence rates, to estimate the value within an error margin $\epsilon$ with high confidence, the required number of rollouts $m$ scales linearly with the variance: $m \propto \text{Var}[R]/\epsilon^2$. Consequently, stochastic actions necessitate repeated sampling (large $m$) to distinguish high-quality nodes from noise, consuming significant NFE for value estimation rather than exploration.

In contrast, multi-model UMF employs *deterministic* actions ($T \approx 0$) using heterogeneous models. In this setting, the transition is deterministic given the action, implying $\text{Var}[R] \approx 0$. Thus, a single rollout ($m = 1$) is sufficient to obtain the exact value of the branch. This design choice yields two practical advantages: (1) **Exploration Width:** By eliminating the need for repeated averaging, UMF can allocate its NFE budget to expand the search tree. (2) **Effective Caching:** Deterministic trajectories allow for aggressive node caching (Algorithm 1). If a state-action pair is revisited, the computation can be skipped entirely. This effectively reduces the marginal cost of exploring known regions of a search tree to zero.

Our empirical results in Table 4 support this analysis. While increasing temperature improves performance (confirming the benefit of kernel diversity from Sec. 5.1), it is consistently outperformed by the multi-model approach. We attribute this to the fact that the multi-model strategy achieves exploration of distinct modes without introducing the variance penalty, maximizing the efficiency of inference-time compute.

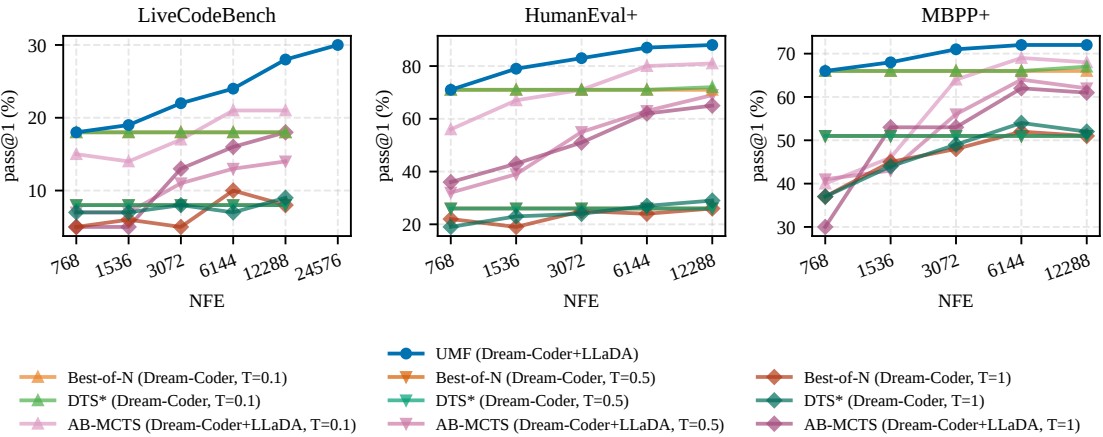

*Figure 2.* Scaling plots (Pass@1) on LiveCodeBench, HumanEval+, and MBPP+.

# 6. Experiments

## 6.1. Experimental Setup

### 6.1.1. BENCHMARKS

We evaluated our approach on coding tasks using 100 samples each from LiveCodeBench (Jain et al., 2025a), HumanEval+, and MBPP+ (via EvalPlus (Liu et al., 2023)). To compute the reward signal during the search, we utilized the public test cases for LiveCodeBench and the standard test cases for EvalPlus, defining the reward as the proportion of passed tests. Candidates with the highest reward were selected for final evaluation. We report the Pass@1 score computed on the private test set (LiveCodeBench) and the extended test set (EvalPlus).

Beyond coding, we also evaluated UMF on the MATH dataset (Hendrycks et al., 2021). We sampled 15 problems from each of the 7 categories, totaling 105 problems. We employed Qwen2.5-Math-PRM-7B (Zhang et al., 2025) to calculate the reward, and used `math_verify` for answer extraction and equivalence checking when computing Pass@1.

### 6.1.2. MODELS

For coding tasks, we employed Dream-Coder-v0-Instruct-7B (Xie et al., 2025) and LLaDA-8B-Instruct (Nie et al., 2025). For math tasks, we used LLaDA-8B-Instruct and Dream-v0-Instruct-7B (Ye et al., 2025). For the Dream model, we set the temperature $T = 0.1$ and utilized the entropy-based remasking strategy. For LLaDA, we set $T = 0$ and used the low-confidence strategy. Our main experiments use standard masked diffusion decoding rather than block diffusion, in order to isolate the effect of non-autoregressive unmasking trajectory search. In Appendix A.2, we show that UMF is also compatible with block diffusion decoding, improving LiveCodeBench Pass@1 from 19.0% to 31.0%.

### 6.1.3. BASELINES

To empirically demonstrate the efficacy of UMF, we compare it against representative inference-time scaling methods ranging from standard Best-of-N (BoN) to advanced tree search algorithms, specifically diffusion-specific tree search (DTS* (Jain et al., 2025b)) and generic adaptive branching MCTS (AB-MCTS-M (Inoue et al., 2025)). This enables a controlled comparison under a matched NFE budget. We evaluated two primary configurations: (1) varying temperature ($T \in \{0.1, 0.5, 1.0\}$) with the deterministic remasking strategy (entropy and low-confidence); and (2) randomized remasking strategy at low temperature ($T \approx 0$). Under these configurations, we tested (A) Best-of-N and (B) DTS* using both Dream and LLaDA models. Additionally, to assess the benefits of multi-model budget allocation, we included a (C) "Pair" baseline that selects the higher-reward solution from independent Dream and LLaDA generations. We also compared against AB-MCTS with two distinct action spaces: (i) a multi-model setting utilizing both Dream-Coder and LLaDA unmasking actions, and (ii) a single-model setting using only Dream-Coder actions. Both configurations were evaluated with temperatures $T \in \{0.1, 0.5, 1.0\}$. In total, these $16 (= (3+1)*2*2)$ ($T$/remask, algorithm, MDLMs) + 6 (pair) + 6 (AB-MCTS) baselines were evaluated across the three coding benchmarks.

For all experiments, we fixed the generation length to 768 tokens and adopted a schedule where one token is unmasked per function evaluation. This mitigates performance degradation from reducing generation length or unmasking multiple tokens simultaneously. Furthermore, to prevent premature padding generation at high temperatures, we applied an EoS padding penalty of $1e{-}12$ for Dream models ($T = 1$) (Xie et al., 2025), and set the confidence of EoS tokens to 0 for LLaDA (Nie et al., 2025), as recommended.

*Table 2.* Pass@1 of UMF on 105 problems from the MATH dataset. Models: Dream-v0-Instruct-7B and LLaDA-8B-Instruct. Reward Model: Qwen2.5-Math-PRM-7B.

| NFE | 768 | 1536 | 3072 | 6144 | 12288 |
|---|---|---|---|---|---|
| Pass@1 (%) | 49.52 | 52.38 | 53.33 | 59.05 | **60.95** |

*Table 3.* LiveCodeBench Pass@1 (%) of UMF with and without cache. The last column reports the cache hit rate; numbers in parentheses denote NFEs saved relative to uncached rollout expansion attempts. Saved NFEs can exceed the executed NFE budget because they are counted relative to the hypothetical uncached expansion attempts.

| NFE | without cache | with cache | Cache Hit Rate (%) (Saved NFEs) |
|---|---|---|---|
| 768 | 18.0 | 18.0 | 0.0 (0) |
| 1536 | 19.0 | 19.0 | 0.0 (0) |
| 3072 | 21.0 | **22.0** (↑ 4.76%) | 47.8 (2108) |
| 6144 | 23.0 | **24.0** (↑ 4.35%) | 54.5 (6375) |
| 12288 | 26.0 | **28.0** (↑ 7.69%) | 55.8 (14186) |

## 6.2. Results

### 6.2.1. RESULTS AT FIXED NFE

Table 1 presents the performance comparison on coding tasks at a fixed budget of $NFE = 12288$. For non-UMF baselines, we report the best result among temperatures $T \in \{0.1, 0.5, 1.0\}$. As observed, UMF consistently outperforms all other tree-search and scaling baselines. Notably, UMF significantly outperforms the "Pair" baselines, which simply split the budget between Dream-Coder and LLaDA. This result confirms that mere access to multiple models is insufficient; the structured interaction provided by the tree search is essential for unlocking their combined potential. Crucially, the results show that strategies relying on random remasking degrade performance. Table 2 further shows the results for UMF on the MATH dataset. UMF maintains high performance at low budgets and also achieves an 11.43 point improvement at $NFE = 12288$. This suggests that UMF generalizes to reasoning tasks beyond coding, provided a valid reward signal exists.

### 6.2.2. PERFORMANCE SCALING WITH TEMPERATURE AND NFE

To analyze scaling behavior, Figure 2 illustrates performance curves for representative methods (full results for all the baselines in Appendix A). The plots reveal that UMF leverages the stability of low-temperature generation in low-budget regimes, while consistently improving performance as the budget increases. To test for saturation, we extended the LiveCodeBench evaluation to $NFE = 24576$, achieving a Pass@1 score of 30.0% (+2.0 points). This continued improvement validates that UMF effectively utilizes the compute budget through mechanisms such as caching. No-

*Table 4.* UMF Pass@1 (%) at NFE=12288 for various action types.

| Action Type | LiveCodeBench | HumanEval+ | MBPP+ |
|---|---|---|---|
| temperature $(T = 0.1, 0.5)$ | 24.0 | 79.0 | 69.0 |
| temperature $(T = 0.1, 1.0)$ | 27.0 | 82.0 | 71.0 |
| remask (entropy, origin) | 20.0 | 82.0 | 68.0 |
| model (Dream-Coder, LLaDA) | **28.0** | **88.0** | **72.0** |

tably, because UMF relies on nearly deterministic unmasking and UCT-based search, its performance scales stably without the fluctuations observed in stochastic baselines.

In contrast, while higher temperatures in baselines enhance diversity, they fail to surpass the $T = 0.1$ setting at $NFE = 12288$, suggesting that the quality loss from stochasticity outweighs the diversity benefits. The only baseline outperforming the deterministic Best-of-N ($T = 0.1$) was AB-MCTS ($T = 0.1$) using multiple models. This implies that exploring distinct modes via model diversity is superior to stochastic perturbation. UMF improves upon AB-MCTS by restricting branching to available actions and optimizing NFE usage.

## 6.3. Ablation Study

### 6.3.1. EFFECTIVENESS OF CACHING IN UMF

To quantify the benefit of caching, we evaluated UMF on LiveCodeBench with and without the caching mechanism. Table 3 shows the results and the cache hit rate, defined as the number of rollouts with cache hits divided by the total number of rollouts. We observe that for $NFE \geq 3072$, the method maintains a high cache hit rate around 50%, and the caching improves the performance for fixed NFE. This demonstrates that caching enables deeper search within the same NFE budget, leading to consistent performance improvements.

### 6.3.2. TYPES OF ACTIONS

We investigated the impact of different action definitions by comparing the multi-model approach against: (1) Temperature scaling ($T = 0.1, 0.5$), (2) Temperature scaling ($T = 0.1, 1.0$), and (3) Remasking strategies (entropy vs. origin). We used Dream-Coder for those baselines. Table 4 presents the results at $NFE = 12288$. Notably, the best single-model UMF variant in Table 4 achieves a 60.0% average Pass@1 across the three coding benchmarks and outperforms the multi-model BoN Pair, DTS* Pair, and AB-MCTS baselines in Table 1 on each benchmark. This indicates that UMF's gains are not merely due to model ensembling. The multi-model configuration further yields the strongest overall results, confirming the additional benefit of structural diversity from heterogeneous models.

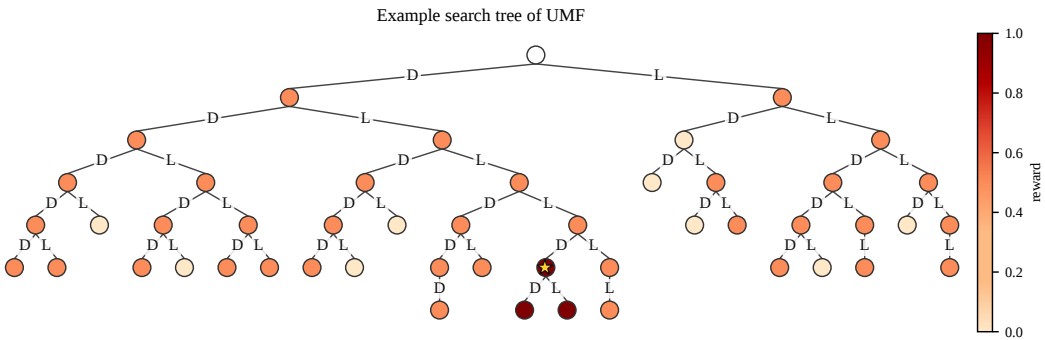

*Figure 3.* Example of the UMF search tree for LiveCodeBench at NFE=12288. "D" denotes unmasking by Dream-Coder, and "L" denotes unmasking by LLaDA. The starred node represents the node used for submission, which is correct for this problem.

```
  1: Dream   2: LLaDA   3: LLaDA   4: LLaDA   5: Dream   <mask>

To solve this problem, we need to determine if we can transform string `S` into
string `T` using the given operations A and B. The operations allow us to swap
segments of `S` and `T` of length `X` and `Y` respectively, under certain conditions.

Here's a step-by-step approach to solve the problem:

1. **Initial Check**: If `S` and `T` are already identical, the answer is "Yes".
2. **Segment Matching**: We need to check if there are segments of length `X` and `Y`
   in `S` and `T` respectively that can be swapped to make `S` equal to `T`.
3. **Transformation Operations**: If such segments exist, we can apply the operations
   A and B to transform `S` into `T`.

Here's the Python code to implement this approach:

```python
def can_transform(N, X, Y, S, T):
    if S == T:
        return "Yes"

    # Check if there are segments of length X and Y in S and T that can be swapped
    for i in range(N - (X + Y) + 1):
        # Check for Operation A
        if S[i:i+X] == '0' * X and S[i+X:i+X+Y] == '1' * Y:
            if T[i:i+X] == '1' * X and T[i+X:i+X+Y] == '0' * Y:
                return "<mask>*382.
3. **Main Function**: The main function reads the input, calls the `can_transform`
function, and prints the result.

This approach ensures that we efficiently determine if it's possible to transform `S`
into `T` using the given operations.<mask>*2
```

*Figure 4.* Text generated by unmasking up to the starred node in Figure 3. Darker colors indicate tokens unmasked earlier. Blue highlights tokens unmasked by Dream-Coder, and orange by LLaDA. The unmasking process proceeded in the order of the numbers shown in the legend.

*Table 5.* Comparison between independent single-model UMFs, two-model UMF, and three-model UMF at NFE=12288. The three-model setup additionally uses DiffuCoder-cpGRPO.

| Method | LiveCodeBench | HumanEval+ | MBPP+ |
|---|---|---|---|
| Pair of 1-model UMFs | 24.0 | 78.0 | 69.0 |
| UMF (2 models) | 28.0 | **88.0** | 72.0 |
| UMF (3 models) | **32.0** | 87.0 | **76.0** |

### 6.3.3. EFFECT OF MULTIPLE MODELS

We aimed to decouple the benefit of multiple models contributing to one trajectory from the benefit of simply allocating budget across two independent models. We compared multi-model UMF against a "Pair" baseline where Dream-Coder and LLaDA solve each problem independently using single-model UMF ($NFE = 6144$ each, total 12288), and the best answer is selected. For the single-model UMF, we used the best-performing UMF configurations from Table 4. Table 5 demonstrates that two-model UMF consistently outperforms the independent pair baseline. Since both methods use the same underlying models and the same total NFE budget, this improvement cannot be explained by simply allocating compute across multiple models. This result highlights the importance of interleaving model capabilities within a single unmasking trajectory. We also evaluate a three-model UMF setup by adding DiffuCoder-cpGRPO (Gong et al., 2025) as an additional action. The average coding score improves from 62.7 to 65.0, suggesting that additional model diversity can further enrich the structural search space explored by UMF.

### 6.4. Case Study: Search and Unmasking Trajectory

To provide qualitative insights into the search process, we focus on a LiveCodeBench problem that was solved only after scaling to $NFE = 12288$. Figure 3 presents the corresponding search tree, while Figure 4 visualizes how the unmasking of the best solution proceeded. The tree structure reveals that the successful trajectory involves interleaved unmasking actions by both Dream-Coder and LLaDA.

Figure 4 further breaks down this trajectory, highlighting the contribution of each model. In this instance, Dream-Coder initiates the solution by outlining implementation steps. LLaDA then fills in specific requirements at the end of the code. Subsequently, LLaDA begins the core implementation, which is finally refined and completed by Dream-Coder. This interplay demonstrates that UMF enables a form of "collaborative generation", where heterogeneous models complement each other to construct a correct solution.

# 7. Conclusion

In this work, we presented UnMaskFork (UMF), a principled test-time scaling framework tailored for Masked Diffusion Language Models. We identified that standard stochastic scaling methods, such as temperature sampling, often degrade the generation quality of MDLMs by disrupting the iterative unmasking trajectory. To address this, UMF replaces stochastic perturbations with deterministic actions that branch the search space using heterogeneous models or distinct inference heuristics. By strictly adhering to deterministic transitions, our method ensures that every explored path retains the high generation quality inherent to the model's optimal decoding process, while deriving diversity from distinct inference configurations. Crucially, this deterministic nature facilitates aggressive node caching, which improves sample efficiency under fixed compute budgets.

Extensive evaluations on coding tasks demonstrate that UMF consistently outperforms existing baselines, including Best-of-N and Diffusion Tree Sampling. Additionally, results on MATH indicate that UMF remains effective for reasoning tasks other than coding. Our findings suggest that maintaining high-quality deterministic backbones and leveraging diverse model priors through interleaved search is more effective than independent ensembling or introducing random noise for inference scaling in non-autoregressive models.

Future work includes training a policy network or value function to guide the tree search, moving beyond heuristic-based UCT for even greater sample efficiency. Furthermore, exploring dynamic action spaces, where the set of candidate models or unmasking schedules adapts to the instance difficulty, could further optimize the trade-off between computational cost and reasoning depth.

## Impact Statement

This paper proposes UnMaskFork (UMF), a test-time scaling method for masked diffusion language models that uses Monte Carlo Tree Search to explore and cache deterministic partial-unmasking trajectories, improving accuracy on code generation and mathematical reasoning tasks under a fixed inference budget. If adopted, UMF could increase the reliability of diffusion-based language models in high-stakes, correctness-sensitive settings (e.g., programming assistance, formalized problem solving, and scientific computing workflows), potentially reducing the need to train larger models or perform additional fine-tuning to reach a target performance level.

At the same time, stronger and more reliable code generation can be misused to produce malicious software, exploit code, or to automate cyberattacks, and more capable reasoning systems can be applied to harmful objectives when paired with an external reward signal. In addition, test-time scaling increases inference-time computation, which may raise energy consumption, latency, and cost, potentially widening access disparities between users with different compute resources. UMF inherits the limitations and biases of the underlying pretrained models, and improved search may amplify undesirable behaviors if the base models are unsafe or poorly calibrated.

We recommend that deployments of UMF follow standard safety practices for generative code and reasoning systems: restrict tool access and execution, sandbox and scan generated code (e.g., static analysis and security review), apply policy and content filters, and use logging and rate limiting. To address environmental and access concerns, practitioners should report inference budgets and consider compute-efficient configurations (including caching) and evaluation on safety-oriented benchmarks alongside capability gains. Our work does not introduce new data collection involving human subjects; the primary societal risks arise from downstream use and deployment choices.

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

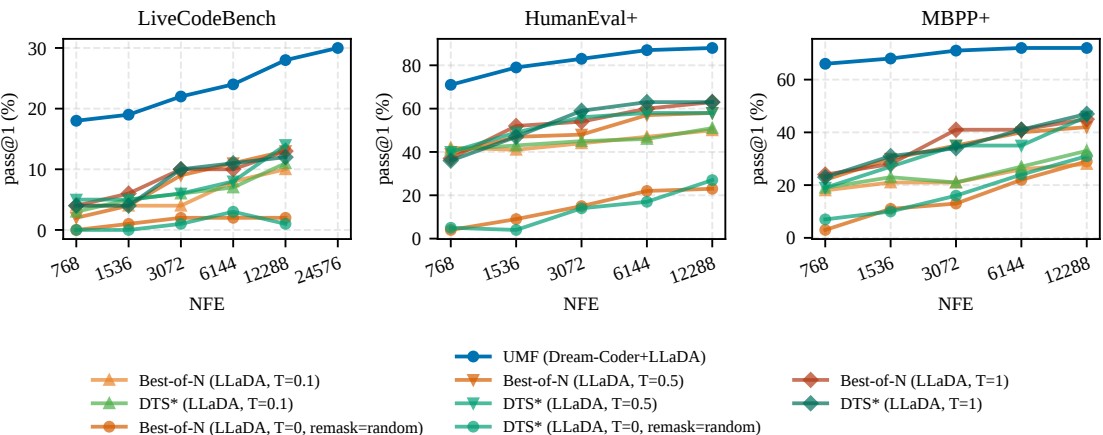

Figure 5. Scaling plots (Pass@1) of LLaDA baselines on LiveCodeBench, HumanEval+, and MBPP+.

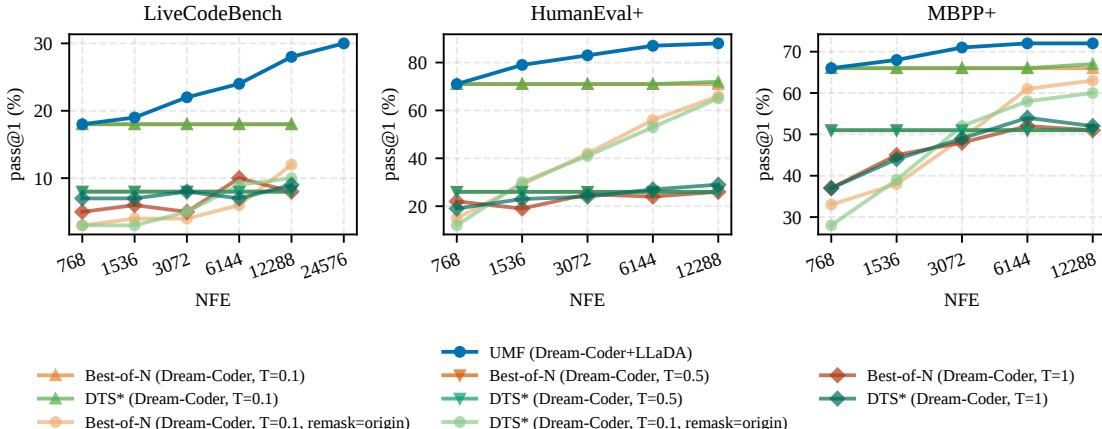

Figure 6. Scaling plots (Pass@1) of Dream baselines on LiveCodeBench, HumanEval+, and MBPP+.

# A. Additional Results

## A.1. Baseline Experiments

In this section, we present comprehensive results on coding tasks for the 28 baseline configurations discussed in Section 6.1.3. The scaling behaviors are illustrated in Figures 5, 6, 7, and 8. Unless otherwise specified, the default remasking strategies are entropy-based and low-confidence strategies.

**LLaDA Baseline TTS methods (Figure 5).** Although the performance improves as the NFE budget increases, UMF consistently outperforms the baseline TTS methods across all benchmarks.

**Dream Baseline TTS methods (Figure 6).** The performance of all TTS baselines remains below that of BoN at $T = 0.1$, which exhibits a nearly flat scaling curve. This plateau in BoN and DTS* supports the observation that Dream-Coder behaves almost deterministically at $T = 0.1$. On LiveCodeBench, all baseline TTS methods underperform compared to deterministic unmasking. Conversely, on HumanEval+ and MBPP+, the origin remasking strategy achieves the most significant performance gain as the NFE budget increases. This highlights the sensitivity of baseline TTS methods to task types, whereas UMF consistently performs best across all tasks.

**Pair Baseline TTS methods (Figure 7).** To calculate the performance of the "Pair" baselines at $NFE = 2k$, we selected the best candidate generated independently by LLaDA and Dream-Coder at $NFE = k$, and then chose the better of the two based on their reward scores. The results demonstrate that UMF outperforms the Pair baselines by a significant margin.

**ABMCTS Baselines (Figure 8).** Leveraging the diversity introduced by distinct MDLMs, multi-model AB-MCTS ($T = 0.1$) at $NFE = 12288$ surpasses the performance of deterministic unmasking. We note that the performance of multi-model

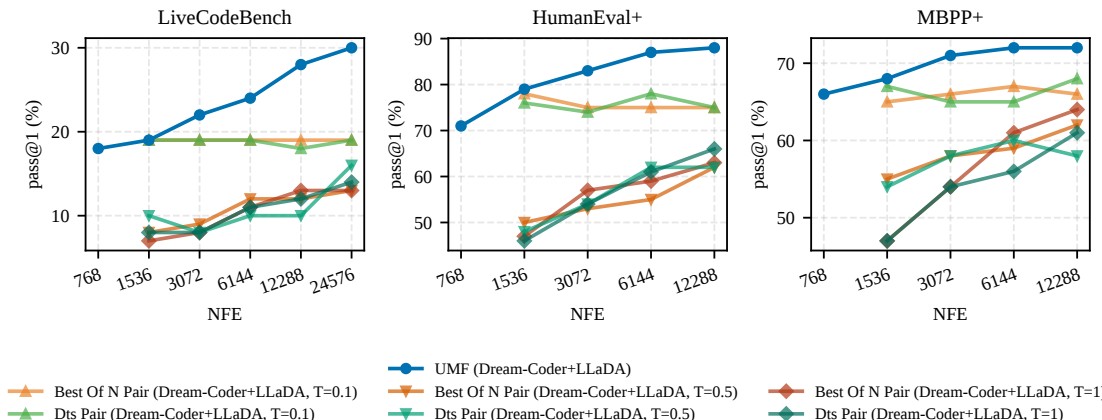

*Figure 7.* Scaling plots (Pass@1) of pair baselines on LiveCodeBench, HumanEval+, and MBPP+.

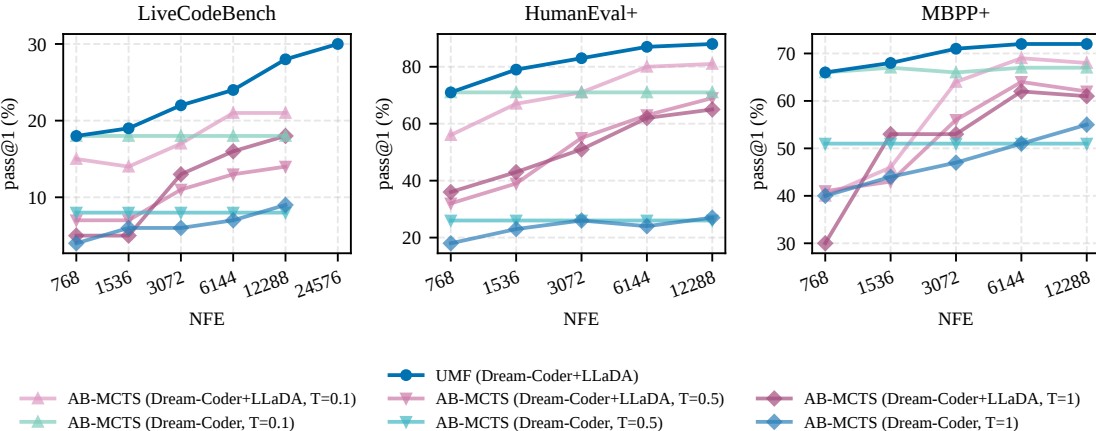

*Figure 8.* Scaling plots (Pass@1) of ABMCTS baselines on LiveCodeBench, HumanEval+, and MBPP+.

*Table 6.* Pass@1 (%) of UMF with block diffusion decoding on LiveCodeBench. Models: Stable-DiffCoder-8B-Instruct and LLaDA-8B-Instruct.

| NFE | 768 | 1536 | 3072 | 6144 | 12288 |
|---|---|---|---|---|---|
| Pass@1 (%) | 19.00 | 18.00 | 23.00 | 28.00 | **31.00** |

*Table 7.* Pass@1 (%) of UMF with two-model and three-model action sets at NFE=12288.

| Method | LiveCodeBench | HumanEval+ | MBPP+ | Average |
|---|---|---|---|---|
| UMF (Dream-Coder, LLaDA) | 28.0 | **88.0** | 72.0 | 62.7 |
| UMF (Dream-Coder, LLaDA, DiffuCoder-cpGRPO) | **32.0** | 87.0 | **76.0** | **65.0** |

AB-MCTS exceeds that of the Pair baselines (Table 1), supporting the claim that collaborative unmasking using distinct MDLMs leads to superior performance.

## A.2. UMF for Block Diffusion MDLMs

To show that UMF can also be applied to block diffusion models, we tested UMF on LiveCodeBench using Stable-DiffCoder-8B-Instruct (Fan et al., 2026), combined with LLaDA-8B-Instruct under block diffusion decoding.

To adapt UMF to block diffusion decoding, we keep the same residual-mask-ratio schedule used in standard UMF, but constrain MCTS branching to block boundaries. Specifically, the current action is held fixed while decoding an active block; UMF does not create a new branch or switch actions inside the block. When a scheduled mask-ratio checkpoint is reached within a block, we defer the MCTS node creation until the block has been fully decoded, and use the resulting block-boundary state as the next search node. This preserves the semi-autoregressive block diffusion procedure while still allowing UMF to branch over deterministic actions along the unmasking trajectory. We use a block size of 4, following the official Stable-DiffCoder configuration.

The results are shown in Table 6. UMF improves LiveCodeBench Pass@1 from 19.0% at 768 NFE to 31.0% at 12288 NFE, a 12-point absolute improvement. These results indicate that UMF is not only compatible with block diffusion decoding, but also provides effective test-time scaling in this semi-autoregressive setting.

## A.3. UMF with Larger Action Sets

To evaluate whether UMF can benefit from larger deterministic action spaces, we extended the two-model action set by adding DiffuCoder-7B-cpGRPO (Gong et al., 2025) as a third action of UMF. The results for the fixed compute budget (12288 NFE) are shown in Table 7.

Table 7 shows that the three-model action set improves the average score from 62.7 to 65.0. Notably, DiffuCoder-cpGRPO is not the strongest standalone coding model among the three MDLMs used in this experiment, yet adding it as an action still improves UMF's overall performance. This suggests that the benefit comes not merely from adding a stronger model, but from increasing the structural diversity of the search space, which allows UMF to discover higher-quality unmasking trajectories.

## A.4. Runtime and Memory Analysis

We also measured wall-clock runtime and memory usage to assess the deployment-level overhead of UMF. Table 8 reports the average per-problem runtime of UMF on 100 LiveCodeBench problems using Dream-Coder and LLaDA on a single H100 GPU. The total runtime scales approximately linearly with the NFE budget, and the dominant cost is MDLM unmasking. In contrast, test execution and search overhead remain comparatively small, while peak GPU memory stays nearly constant across budgets.

The GPU memory in Table 8 corresponds to keeping both MDLMs resident on the GPU. In memory-constrained settings, the inactive model can instead be offloaded to CPU memory. In our measurements, Dream-Coder and LLaDA require 14.2 GiB and 14.9 GiB of GPU memory, respectively, and transferring model weights between pinned CPU memory and GPU took 0.56/0.54 s for Dream-Coder and 0.59/0.57 s for LLaDA, for CPU-to-GPU/GPU-to-CPU transfer. Thus, even at

*Table 8.* Average per-problem runtime and memory usage of UMF (Models: Dream-Coder and LLaDA) on LiveCodeBench. The number of switches in this table counts model swaps across the entire MCTS search for one problem, whereas the tokenizer-switch bound in Section 4.3 is per single unmasking trajectory.

| NFE | Total Time (s) | Unmask Time (s) | Eval Time (s) | CPU RAM (GiB) | GPU RAM (GiB) | # Switches |
|---|---|---|---|---|---|---|
| 768 | 42.74 | 41.56 | 1.17 | 4.32 | 35.64 | 0.0 |
| 1536 | 85.32 | 82.65 | 2.64 | 4.38 | 35.64 | 1.0 |
| 3072 | 175.40 | 164.65 | 10.65 | 4.41 | 35.64 | 2.6 |
| 6144 | 362.99 | 327.54 | 35.21 | 4.39 | 35.64 | 5.7 |
| 12288 | 702.86 | 647.14 | 55.19 | 4.40 | 35.64 | 13.5 |

*Table 9.* Average per-problem runtime of UMF (Models: Dream-v0, LLaDA, PRM: Qwen2.5-Math-PRM-7B) on MATH at NFE=3072.

| Task | Total Time (s) | Unmask Time (s) | PRM Time (s) | # PRM Calls | # MDLM Switches | Peak GPU RAM (GiB) |
|---|---|---|---|---|---|---|
| MATH | 108.93 | 108.48 | 0.30 | 5.00 | 2.38 | 46.9 |

NFE=12288, where UMF performs 13.5 model switches on average, CPU offloading adds only a modest overhead compared with the total unmasking time.

For mathematical reasoning, UMF uses a PRM only to score fully unmasked terminal rollouts, as in standard test-time scaling protocols. Table 9 shows that the PRM overhead is small relative to MDLM unmasking time. This is because the PRM is not invoked at every denoising step, but only for terminal candidate solutions. Moreover, as discussed above, the PRM can be offloaded to CPU memory when it is inactive, thereby avoiding the need to keep the PRM resident on GPU alongside the MDLMs.

### A.5. Robustness to Public-Test Mismatch

To evaluate the robustness of UMF to verifier mismatch, we randomly dropped a fraction of public test cases used as the search reward of UMF on LiveCodeBench, and evaluated Pass@1 on private tests.

As shown in Table 10, UMF maintains the same Pass@1 when 10% of the public tests are removed, and still retains most of its performance even when 50-80% of the public tests are dropped. This suggests that UMF is reasonably robust to moderate reward mismatch between the public tests used during search and the private tests used for final evaluation.

### A.6. MATH Evaluation with Different Answer Extraction Methods

In the main text, we report MATH accuracy using `math_verify`, which provides a stricter answer extraction and equivalence checking pipeline. For completeness, we also report results using a simpler rule-based answer extraction procedure. Both evaluations are applied to the same generated outputs; only the answer extraction and verification pipeline differs.

As shown in Table 11, the two evaluation procedures yield the same overall trend: UMF improves as the NFE budget increases. The `math_verify` pipeline gives slightly higher scores in our evaluation, but the scaling behavior is consistent across both extraction methods.

### A.7. Comparison with Remasking and Search Baselines

We additionally compare UMF with recent inference-time scaling methods for masked diffusion language models, ReMDM (Wang et al., 2025) and TReASURe (Yu et al., 2025). Since these methods target different search formulations and reward interfaces, the comparisons should be viewed as additional empirical references rather than exact drop-in replacements for our main baselines. We used the authors' recommended settings where applicable and matched NFE according to MDLM forward passes.

For coding tasks, we compare against ReMDM on LiveCodeBench. ReMDM revisits and remasks previously decoded tokens, whereas UMF explores different deterministic unmasking actions and evaluates fully unmasked terminal candidates using execution-based rewards.

*Table 10.* Pass@1 (%) under public-test mismatch on LiveCodeBench at NFE=12288.

| Public test drop rate | 0.0 | 0.1 | 0.5 | 0.8 |
|---|---|---|---|---|
| Pass@1 (%) | 28.00 | 28.00 | 25.00 | 24.00 |

*Table 11.* MATH Pass@1 (%) of UMF under two answer extraction methods.

| NFE | Rule-based | `math_verify` |
|---|---|---|
| 768 | 45.71 | 49.52 |
| 1536 | 48.57 | 52.38 |
| 3072 | 49.52 | 53.33 |
| 6144 | 54.29 | 59.05 |
| 12288 | 58.10 | 60.95 |

Table 12 shows that ReMDM does not improve with additional NFE on LiveCodeBench in our setting, while UMF continues to scale. This suggests that localized remasking alone may be insufficient for producing the structural changes needed for coding problems, whereas UMF can explore more diverse unmasking trajectories through deterministic action branching.

For mathematical reasoning, we compare against TReASURe on MATH, where PRM-based scoring makes reward-guided search feasible. TReASURe uses step-wise reward estimates on intermediate masked states, while UMF evaluates terminal rollouts only.

Table 13 shows that UMF substantially outperforms TReASURe in this setting. We hypothesize that, for math reasoning, rewards computed on partially masked intermediate states are much noisier than rewards computed on fully unmasked solutions. By evaluating only terminal candidates, UMF avoids this step-wise reward mismatch while still using tree search to allocate inference compute.

*Table 12.* LiveCodeBench Pass@1 (%) comparison between ReMDM and UMF under matched NFE budgets.

| Method | 1536 | 3072 | 6144 | 12288 |
|--------|------|------|------|-------|
| ReMDM | 18.0 | 18.0 | 18.0 | 18.0 |
| UMF | **19.0** | **22.0** | **24.0** | **28.0** |

*Table 13.* MATH Pass@1 (%) comparison between TReASURe and UMF.

| Method | NFE | Rule-based | `math_verify` |
|--------|-----|------------|---------------|
| TReASURe | 1536 | 6.67 | 9.52 |
| TReASURe | 3072 | 6.67 | 7.62 |
| TReASURe | 6144 | 10.48 | 10.48 |
| UMF | 1536 | 48.57 | 52.38 |
| UMF | 3072 | 49.52 | 53.33 |
| UMF | 6144 | 54.29 | 59.05 |

# B. Detailed Remasking Strategies

In this section, we provide the formal definitions for the remasking strategies ($g_a$) discussed in Section 3.5. Let $\text{TopK}(\{v_i\}_{i \in I}, k)$ denote the index set of the $k$ largest values among $\{v_i\}_{i \in I}$.

**Dream Strategies.** These strategies operate globally on the masked indices $\mathcal{M}(z_t)$. Let $\alpha_t \in [0, 1]$ denote the target mask ratio at step $t$.

- **Dream (entropy) [Deterministic]:** Used in UMF. This strategy unmasks positions where the model is most confident. We compute the entropy $H_{t,i} := -\sum_v p_{\theta,i}(v|z_t) \log p_{\theta,i}(v|z_t)$ for each masked position. Defining confidence as $c_{t,i} := -H_{t,i}$, and letting $k_t$ be the number of tokens to unmask, the commit set is:

$$S_t := \text{TopK}(\{c_{t,i}\}_{i \in \mathcal{M}(z_t)}, \ k_t).$$

- **Dream (origin) [Stochastic]:** Used as a baseline. This strategy employs independent probabilistic unmasking. With transition probability $p_t := 1 - \alpha_{t-1}/\alpha_t$, we sample $b_i \sim \text{Bernoulli}(p_t)$ for each $i \in \mathcal{M}(z_t)$ and set $S_t := \{i \mid b_i = 1\}$.

**LLaDA Strategies.** These strategies are applicable to both block-based and standard diffusion settings. For generality, we formulate them with respect to an active generation block $B$; for standard decoding, $B$ simply encompasses the entire sequence. Let $\mathcal{M}^B(z_t) := \mathcal{M}(z_t) \cap B$ denote the masked indices within the block.

- **LLaDA (low-confidence) [Deterministic]:** Used in UMF. We define the confidence score $c_{t,i} := p_{\theta,i}(\hat{x}_i \mid z_t)$ as the probability of the proposed token $\hat{x}_i$. The commit set selects the top-$k_t$ most confident tokens:

$$S_t := \text{TopK}(\{c_{t,i}\}_{i \in \mathcal{M}^B(z_t)}, \ k_t).$$

- **LLaDA (random) [Stochastic]:** Used as a baseline. We draw independent scores $u_i \sim \text{Uniform}(0, 1)$ and set $S_t := \text{TopK}(\{u_i\}_{i \in \mathcal{M}^B(z_t)}, \ k_t)$, which corresponds to uniform random selection.

