# OpenReview forum: "UnMaskFork: Test-Time Scaling for Masked Diffusion via Deterministic Action Branching"
_ICML.cc/2026/Conference — ICML 2026 regular_

### Official Review · Reviewer_Nuc1 · 2026-02-24

**Soundness:** 2
**Presentation:** 3
**Significance:** 3
**Originality:** 3
**Overall Recommendation:** 3
**Confidence:** 4

**Summary:**

The paper frames masked diffusion decoding as MCTS over unmasking trajectories, with deterministic action branching and cache reuse (Section 4.2/4.3). The topic is important for inference-time scaling in diffusion models and is connected to DTS/AB-MCTS style search lines.[1][2]

**Compliance With Llm Reviewing Policy:**

Affirmed.

**Key Questions For Authors:**

1. In Section 6.2, can you isolate UMF under single-model action spaces to separate algorithmic gain from ensemble gain?
2. Can you add matched wall-clock and peak-memory curves alongside NFE curves?
3. Can you add direct comparisons with remasking/search alternatives under equal compute budgets?[3][4]
4. How robust is UMF when reward models are partially noisy or weakly aligned?

[1] Jain et al. *Diffusion Tree Sampling: Scalable inference-time alignment of diffusion models*. OpenReview (NeurIPS 2025). https://openreview.net/forum?id=3D88hCO0Gd

[2] Inoue et al. *Wider or Deeper? Scaling LLM Inference-Time Compute with Adaptive Branching Tree Search (AB-MCTS)*. arXiv:2503.04412. https://arxiv.org/abs/2503.04412

[3] Wang et al. *Remasking Discrete Diffusion Models with Inference-Time Scaling*. arXiv:2503.00307. https://arxiv.org/abs/2503.00307

[4] Yu et al. *Tree Reward-Aligned Search for TReASURe in Masked Diffusion Language Models*. arXiv:2509.23146. https://arxiv.org/abs/2509.23146

[5] Sahoo et al. *Simple and Effective Masked Diffusion Language Models*. arXiv:2406.07524. https://arxiv.org/abs/2406.07524

**Limitations:**

`No.` The paper needs stronger analysis on reward-model robustness and deployment-level compute accounting.

**Strengths And Weaknesses:**

### Strengths
1. Section 4.1 motivation is technically reasonable: stochastic perturbations can destabilize masked diffusion trajectories.[1][5]
2. The method in Section 4.2 is practical, and cache reuse is integrated tightly into the search loop.
3. Section 6.2 reports competitive coding results under fixed NFE budgets.

### Weaknesses
1. In Section 4.3, the action set includes model switching. Then Section 6.2 mainly compares against single-model baselines, which mixes algorithmic gain with multi-model ensemble gain.
2. Section 5.2 and Section 6 emphasize NFE efficiency, but deployment relevance also needs wall-clock and memory accounting.
3. The deterministic design in Section 5.2 likely improves stability, but diversity trade-offs are under-analyzed versus remasking/search alternatives.[3][4]
4. Section 6.2 performance depends on reward quality; robustness under noisy/misaligned verifiers is not analyzed deeply enough.[1]

---

> ### Author Rebuttal · Authors · 2026-03-31
>
> We thank the reviewer for their feedback. Below, we address the concerns and summarize the updates made in our current draft.
>
> ---
> ## W1/Q1: Separating Algorithmic Gain from Ensemble Gain
> To isolate algorithmic gain, Table 4 evaluates single-model UMF with Dream-Coder only (actions: temperature or remasking). At NFE=12288, it reaches 27.0% Pass@1 on LiveCodeBench, versus 18.0% for single-model Best-of-N and DTS* under the same budget.
>
> To isolate simple ensembling effects, we added a “Pair” baseline that splits the budget across independent models and keeps the higher-reward output. Table 5 shows that multi-model UMF (28.0%) outperforms a Pair of independent single-model UMFs (24.0%). This suggests that the gain is not from simple ensembling, but from allowing multiple MDLMs to alternately unmask the trajectory, yielding more diverse search paths.
>
> ## W2/Q2: Runtime and Memory
> We measured wall-clock time for 100 LiveCodeBench problems using UMF (Dream-Coder + LLaDA) on a single H100 GPU, tracking MDLM forward passes ("Unmask Time") and test execution ("Eval Time"):
>
> |NFE|Total Time(s)|Unmask Time(s)|Eval Time(s)|RAM(GiB)|GPU RAM(GiB)|# MDLM Switches|
> |-|-|-|-|-|-|-|
> |768|42.74|41.56|1.17|4.32|35.64|0.0|
> |1536|85.32|82.65|2.64|4.38|35.64|1.0|
> |3072|175.4|164.65|10.65|4.41|35.64|2.6|
> |6144|362.99|327.54|35.21|4.39|35.64|5.7|
> |12288|702.86|647.14|55.19|4.4|35.64|13.5|
>
> Total time scales roughly linearly with NFE, while RAM/GPU RAM remain nearly constant. Unmasking dominates runtime, with evaluation and search overhead comparatively small.
>
> CPU offloading of the inactive model saves about 15 GiB GPU RAM (Dream-Coder 14.2 GiB; LLaDA 14.9 GiB); transfer times are 0.56/0.54 s for Dream-Coder and 0.59/0.57 s for LLaDA (CPU→GPU / GPU→CPU), so a full swap takes about 1.2 s. Even at NFE=12288 (13.5 switches), this adds only ~2.5% runtime overhead, so multi-model UMF can approach single-model GPU RAM usage with modest overhead.
>
> ## W3: UMF Quality/Diversity
> Unlike [3] and [4], UMF does not rely on stochastic sampling for diversity; instead, it generates diverse candidates by branching partially unmasked sequences with different MDLM actions.
>
> We evaluated the answer candidate codes generated by UMF (Dream-Coder+LLaDA) on LiveCodeBench. We measure diversity by normalized AST edit distance and quality by mean public test score:
>
> |NFE|Quality (mean public score)|Diversity (mean normalized AST edit distance)|
> |-|-|-|
> |768|0.406|0.000|
> |1536|0.440|0.370|
> |3072|0.541|0.277|
> |6144|0.604|0.287|
> |12288|0.701|0.287|
>
> As NFE increases, the mean AST distance remains substantial rather than collapsing, while the mean public score steadily improves. This suggests that newly generated candidates remain structurally distinct from earlier ones, and that MCTS continues to prioritize higher-quality solutions during search.
>
> ## W4/Q4: Robustness under Noisy Rewards
> As in [1], UMF computes rewards only on fully rolled-out outputs, since partially unmasked states in coding and math are not meaningful final solutions.
>
> To evaluate UMF's robustness against noisy rewards, we simulated a noisy environment by injecting Gaussian noise ($N(0, σ^2)$) into the LiveCodeBench reward (defined as the proportion of passed tests, $r\in [0, 1]$). We then measured the performance of UMF (Dream-Coder and LLaDA) under varying noise intensities:
>
> |σ|768|1536|3072|6144|12288|
> |-|-|-|-|-|-|
> |0.0|18.0|19.0|22.0|24.0|28.0|
> |0.1|18.0|19.0|22.0|25.0|28.0|
> |0.5|18.0|18.0|20.0|23.0|27.0|
> |1.0|18.0|15.0|17.0|17.0|20.0|
>
> The performance of UMF remains stable up to σ=0.5 and degrades at σ=1.0, indicating robustness to moderate reward noise and the expected failure mode at very low signal-to-noise ratio.
>
> ## Q3: Comparison to Remasking/Search Alternatives
> We agree that direct comparisons are valuable. ReMDM [3] and TReASURe [4] target settings different from ours: ReMDM lacks terminal-reward-guided search, while TReASURe relies on step-wise rewards, which are weak/noisy for coding and math and expensive for execution-based tasks such as LiveCodeBench.
>
> We therefore compare UMF with ReMDM on LiveCodeBench, and with TReASURe on MATH, where PRM-based evaluation makes the latter feasible.
>
> ReMDM LiveCodeBench Pass@1 vs. UMF Pass@1
> |NFE|Pass@1(%)|UMF Pass@1 (%)|
> |-|-|-|
> |1536|18.0|19.0|
> |3072|18.0|22.0|
> |6144|18.0|24.0|
> |12288|18.0|28.0|
>
> TReASURe MATH Pass@1 vs. UMF Pass@1
> |NFE|Pass@1(%)|UMF Pass@1 (%)|
> |-|-|-|
> |1536|6.7|48.6|
> |3072|6.7|49.5|
> |6144|10.5|54.3|
>
> ReMDM does not improve with NFE on LiveCodeBench, likely because localized remasking changes surface tokens without structural modification of the generated code. TReASURe also performs poorly on MATH, suggesting that step-wise rewards on incomplete text are too noisy in this setting.
>
> In contrast, UMF scales across both tasks by evaluating only fully unmasked terminal sequences, which provides a more reliable reward signal without step-wise evaluation overhead. We added these comparisons to our current draft.

---

> > ### Author Rebuttal · Reviewer_Nuc1 · 2026-04-05
> >
> > Thank you for the detailed rebuttal. The additional analyses are helpful, and they address part of my concerns. However, I am maintaining my score at 3.
> >
> > In particular, I still do not think the current evidence fully separates the algorithmic benefit of UMF from the benefit of heterogeneous multi-model action choices, nor does it provide a sufficiently matched deployment-level comparison under equal wall-clock and memory budgets. I also find the robustness analysis only partially convincing: synthetic reward noise is not a full substitute for realistic verifier misalignment, and the added comparisons to related remasking/search methods are useful but not fully apples-to-apples.  More broadly, my view remains that the contribution is technically interesting but somewhat incremental relative to recent diffusion/tree-search work. For these reasons, I believe weak reject is still the most appropriate recommendation from my side.

---

> > > ### Author Response · Authors · 2026-04-06
> > >
> > > We sincerely thank you for the thoughtful feedback. We address the remaining concerns below.
> > >
> > > ## Algorithmic Benefit of UMF
> > > To isolate algorithmic gain from ensembling:
> > >
> > > 1. **1-model UMF beats single-model baselines**: At matched NFE, Tables A–C below show UMF outperforms baselines even with a single MDLM. Thus, the gains cannot be explained solely by multi-model choices.
> > > 1. **1-model UMF beats baselines enhanced with model ensembling**: Table A shows that 1-model UMF (60.0) outperforms the BoN pair (53.3), DTS pair (53.7), and multi-model AB-MCTS (56.7). Thus, UMF’s core algorithm is stronger even when these baselines benefit from multi-model effects.
> > > 1. **Multi-model UMF outperforms independent 1-model UMF (pair)**: Table A shows that 1-model UMF (pair) (57.0) is outperformed by 2-model UMF (62.7). This indicates within-trajectory collaboration, not independent ensembling.
> > >
> > > |Algorithm|Pass@1 (NFE=12288)|
> > > |-|-|
> > > |BoN|51.7|
> > > |BoN (pair)|53.3|
> > > |DTS|52.3|
> > > |DTS (pair)|53.7|
> > > |multi-model AB-MCTS|56.7|
> > > |**1-model UMF**|**60.0**|
> > > |1-model UMF (pair)|57.0|
> > > |**2-model UMF**|**62.7**|
> > > |**3-model UMF**|**65.0**|
> > >
> > > Table A: Avg. Pass@1 (%) across three coding tasks.
> > >
> > > |Algorithm|Pass@1 (NFE=12288)|
> > > |-|-|
> > > |ReMDM|18.0|
> > > |**1-model UMF**|**27.0**|
> > > |**2-model UMF**|**28.0**|
> > > |**3-model UMF**|**32.0**|
> > >
> > > Table B: LiveCodeBench Pass@1 (%).
> > >
> > > |Algorithm|Pass@1 (NFE=6144)|
> > > |-|-|
> > > |TReASURe|10.5|
> > > |**1-model UMF**|**50.5**|
> > >
> > > Table C: MATH Pass@1 (%); 1-model UMF newly added (T=0.1/1.0).
> > >
> > > ## Matched Deployment-Level Comparison
> > > Our main claim concerns performance at matched NFE budgets, following the standard protocol in the literature, including DTS/TReASURe, because NFE isolates MDLM forward compute. We also measured runtime/memory. These results show inactive-model CPU offloading makes multi-model UMF practical, while 1-model UMF already outperforms matched single-model baselines under the same single-model memory footprint.
> > >
> > > ## Robustness to Verifier Misalignment
> > > To test a practical verifier mismatch, we dropped a fraction of public test cases, increasing mismatch between search reward and private Pass@1.
> > > |public test drop rate|Pass@1 (NFE=12288)|
> > > |-|-|
> > > |0|28.0|
> > > |0.1|28.0|
> > > |0.5|25.0|
> > > |0.8|24.0|
> > >
> > > Even under severe misalignment (80% drop), UMF maintains performance, showing robustness to a realistic verifier misalignment.
> > >
> > > Unlike our code and math setting, prior work evaluates tree search on different task families: DTS on class-conditional/image generation and language completion, and TReASURe on perplexity, linguistic acceptability, and sentiment/toxicity control. In coding and math, partially unmasked states are weak surrogates for final correctness, so reliable verification requires fully unmasked solutions.
> > >
> > > UMF avoids this by evaluating only terminal rollouts. In coding tasks, the main noisy-reward issue is the mismatch between public-test reward and private-test correctness on terminal solutions, which UMF handles well.
> > > ## Comparisons to Remasking/Search Methods
> > > For the algorithmic claim, all methods use a matched NFE budget, the standard protocol for inference-time scaling. This matches the dominant MDLM forward-pass compute in UMF.
> > >
> > > Also, TReASURe is wall-clock inefficient. At NFE=768 on LiveCodeBench, TReASURe takes 1490.83s/problem vs. 42.74s/problem for UMF. Since MDLM forward-pass time is nearly identical, the gap comes almost entirely from reward evaluation at each masking step (1453.3s), making step-wise search substantially more expensive in our coding setting.
> > > ## Novelty of UMF
> > > UMF’s novelties constitute a distinct test-time-scaling design rather than an incremental update:
> > >
> > > 1. **Expanding the search space while preserving each model’s best deterministic configuration**: Existing methods explore different axes of diversity: ReMDM enables iterative refinement by remasking previously decoded tokens; DTS searches over denoising trajectories through stochastic rollouts; and TReASURe branches over reveal order and token content with deterministic proxy scoring. UMF instead treats distinct models or deterministic heuristics as discrete actions. This expands the search space while strictly maintaining optimal, deterministic unmask configurations, effectively bypassing quality loss by stochastic unmasking.
> > > 1. **Enabling within-trajectory collaborative unmasking**: UMF is not merely benefiting from ensembling. Under the same NFE budget, 2-model UMF outperforms independent pairing of 1-model UMF (62.7 vs. 57.0), showing that the gain comes from within-trajectory collaboration rather than independent selection. UMF can therefore discover beneficial interleavings of MDLMs that are inaccessible to independent ensembling (Figs. 3–4).
> > > 1. **Consistently superior empirical performance**: Tables A–C show that UMF outperforms all baselines. Crucially, 1-model UMF beating ensembled baselines provides evidence of a true algorithmic gain rather than a mere ensemble effect.
> > >
> > > We hope this clarifies our main claims for your final evaluation.

---

### Official Review · Reviewer_9D2q · 2026-03-10

**Soundness:** 3
**Presentation:** 4
**Significance:** 3
**Originality:** 3
**Overall Recommendation:** 4
**Confidence:** 3

**Summary:**

The paper introduces UnMaskFork (UMF), a test-time scaling framework for Masked Diffusion Language Models (MDLMs) that treats the iterative unmasking process as a search problem. Instead of increasing randomness (for example, high-temperature sampling), UMF uses deterministic action branching (choices like model and unmasking heuristic) to build a search tree over partially masked states, applies Monte Carlo Tree Search to allocate inference compute, and uses reward-based evaluation of completed candidates to guide the search; determinism also enables aggressive caching/reuse of intermediate results for better efficiency under a fixed NFE budget. The paper reports improvements on coding and math settings where an evaluation signal (tests or a reward model) is available.

**Compliance With Llm Reviewing Policy:**

Affirmed.

**Final Justification:**

*   The authors meaningfully addressed most of my original questions.
*   However, given that concerns raised by another reviewer (e.g., the lack of matched wall-clock and memory comparisons) were not fully resolved, I decided to maintain my score.

**Key Questions For Authors:**

1. Under the same NFE budget, how much of UMF’s gain is attributable to caching? Please report **cache hit rate / NFE saved**, and include an ablation that disables caching while keeping the same MCTS and action set
2. How does performance change as the action set grows from 2 (Dream + LLaDA) to **3–4 actions** (additional MDLM checkpoints and/or additional deterministic policy/temperature variants), under the same NFE budgets
3. Since UMF stresses low-temperature / deterministic rollouts but uses Dream with (T=0.1), please **include Dream with (T=0)** (and maybe a small sweep around it) under the same UMF budget to show whether the “near-zero temperature” story holds for Dream as well.

**Limitations:**

See Weakness.

**Strengths And Weaknesses:**

### Strengths

1. UMF is formulated as an inference-time search procedure that can, in principle, plug into different pretrained MDLM checkpoints and different unmasking heuristics, and the paper shows gains over single-strategy decoding with increasing NFE budgets.
2. The use of deterministic actions makes intermediate trajectories reusable, allowing aggressive caching to reduce repeated compute under a fixed NFE budget.

### Weakness

1. Experiments mainly instantiate UMF with a small number of action types and two MDLM families; it is unclear how robust the framework is with **larger action sets** (for example, 3–4 models or more policy variants).
2. Since UMF is inference-only (no model training), it would be useful to add a matched-cost comparison against a *slightly stronger* pretrained baseline (for example, a larger or better-tuned checkpoint) decoded with a standard method, to clarify when UMF’s extra inference compute gives a better return than modest model improvements. At the same time, UMF remains significant as a training-free lever for deployment settings where upgrading or retraining checkpoints is costly, and it provides a clear way to trade NFE for accuracy.

---

> ### Author Rebuttal · Authors · 2026-03-31
>
> We thank the reviewer for the constructive feedback and for recognizing UMF's strengths. Below, we address the raised questions and summarize the corresponding updates made in our current draft.
>
> ---
> ## W1/Q2: UMF with Larger Action Sets
> We agree that evaluating UMF with a larger action set is crucial to demonstrating its scalability. To test this, we expanded the 2-model UMF (Dream-Coder + LLaDA) by adding DiffuCoder-7B-cpGRPO as a third action. Results on coding tasks at NFE=12288 are below:
>
> |Method|LiveCodeBench|HumanEval+|MBPP+|Avg.|
> |-|-|-|-|-|
> |3-model UMF|**32.0**|87.0|**76.0**|**65.0**|
> |2-model UMF|28.0|**88.0**|72.0|62.7|
>
> The 3-model UMF outperforms the 2-model setup on average. Notably, although DiffuCoder-7B-cpGRPO is a weaker standalone model with lower reported baseline scores on LiveCodeBench, HumanEval+, and MBPP+, its inclusion still enhances overall performance. This improvement stems from increased structural diversity: collaborative unmasking across three distinct MDLMs enriches the search space, allowing UMF to discover previously inaccessible, higher-scoring paths. Details are included in Appendix A of our current draft.
>
> ## W2: Matched-Cost Comparison to Tuning and Larger Models
> We thank the reviewer for the opportunity to clarify the matched-cost comparison. To compare UMF with task-specific tuning, we evaluated DiffuCoder-7B-cpGRPO, obtained by coupled-GRPO from DiffuCoder-7B-Instruct, against UMF using DiffuCoder-7B-Instruct and LLaDA-8B-Instruct. Average scores across three coding tasks are below:
>
> |NFE|768|1536|3072|6144|12288|
> |-|-|-|-|-|-|
> |DiffuCoder-cpGRPO|46.0|-|-|-|-|
> |DiffuCoder-Instruct (UMF w/ LLaDA-8B)|41.0|43.3|**48.0**|**51.3**|**52.7**|
>
> UMF surpasses the tuned cpGRPO checkpoint at a budget of NFE=3072 and continues to scale. One cpGRPO training run costs approximately $5.75 \times 10^7$ forward-equivalents (21K RL instances, G=10 rollouts, 256 denoising steps, plus update costs). In contrast, UMF at budget $B$ adds only $B-768$ passes per query. This yields a break-even volume of ~$2.5 \times 10^4$ queries at $B=3072$. Thus, UMF provides a superior compute return for low-to-medium volume deployments; tuning only becomes favorable when one-time training costs are amortized over tens of thousands of queries.
>
> To compare UMF with scaling model size, we evaluated our 7B/8B UMF pipeline against larger 16B-class models: LLaDA-2.0-mini and LLaDA-2.1-mini (F and Q modes). We compared the average scores on LiveCodeBench, HumanEval+, and MBPP+ (using official scores for the 16B models):
>
> |Method/Model|Pass@1 (%)|
> |-|-|
> |UMF (Dream-Coder-7B/LLaDA-8B, NFE=12288)|62.67|
> |LLaDA-2.0-mini (16B)|**63.82**|
> |LLaDA-2.1-mini (16B, F)|60.87|
> |LLaDA-2.1-mini (16B, Q)|62.47|
> |Dream-Coder (7B, NFE=768) |51.67|
>
> UMF improves the 7B baseline by 11 points, matching 16B-class performance. Scaling from 7B to 16B typically doubles pre-training compute and inference GPU RAM requirements per request. In this context, UMF’s additional NFE is modest compared to the massive infrastructure costs of training and hosting 16B models. UMF thus offers a cost-effective alternative to reach 16B-level performance using 7B-class models.
>
> ## Q1: Caching Ablation
> In Table 3, we conducted an ablation study where we disabled the cache while maintaining the exact same MCTS configuration and action set (using Dream-Coder and LLaDA). In our current draft, we additionally report the NFE saved. The expanded table is presented below:
>
> UMF Pass@1 (%) on LiveCodeBench with and without cache
> |NFE|w/o cache|w/ cache|Cache Hit Rate (%)|NFE Saved|
> |-|-|-|-|-|
> |768|18.0|18.0|0|0|
> |1536|19.0|19.0|0|0|
> |3072|21.0|22.0|47.8|2108|
> |6144|23.0|24.0|54.5|6375|
> |12288|26.0|28.0|55.8|14186|
>
> As demonstrated, UMF effectively saved NFEs, leading to stable improvements compared to the case without cache.
>
> ## Q3: Dream-Coder with T=0
> We thank the reviewer for raising this important point. We agree that verifying the model's behavior at T=0 is a natural expectation given our emphasis on deterministic rollouts. However, we did not evaluate Dream-Coder at T=0 because it is a known issue that the model's performance severely degrades at this exact temperature setting.
>
> This issue has been reported in the model's official repository (GitHub issues [1] and [2]). According to the developers, setting the temperature to exactly zero causes premature EOS/PAD generation, leading to early stopping and severely truncated trajectories. We also reproduced this failure mode in our preliminary experiments.
>
> Therefore, we used T=0.1 for a fair and functional evaluation. This value is recommended by the original Dream paper and by the developer in [2], and is the closest stable approximation to determinism without causing this issue. We note this in Appendix A of our current draft.
>
> [1] https://github.com/DreamLM/Dream/issues/84
>
> [2] https://github.com/DreamLM/Dream/issues/56

---

> > ### Author Rebuttal · Reviewer_9D2q · 2026-04-03
> >
> > The rebuttal meaningfully addresses most of my original questions, and I appreciate the added evidence on caching, action-set scaling, and matched-cost comparisons. However, after also considering concerns raised by other reviewers, I think keeping my current overall recommendation is the most appropriate decision.

---

> > > ### Author Response · Authors · 2026-04-06
> > >
> > > We are very grateful for your continued engagement and are glad to hear that our initial rebuttal successfully addressed your main questions.
> > >
> > > Regarding the broader limitations raised by other reviewers that you noted in your follow-up, we have posted a more comprehensive breakdown in a new response to Reviewer Nuc1. That response includes clear evidence on single-model algorithmic gains, the benefit of within-trajectory multi-model collaboration over independent pairing, and robustness under practical verifier mismatch. We believe these additional clarifications help sharpen the intended claim of the paper.

---

### Official Review · Reviewer_pbKi · 2026-03-11

**Soundness:** 3
**Presentation:** 4
**Significance:** 3
**Originality:** 3
**Overall Recommendation:** 4
**Confidence:** 4

**Summary:**

This study proposes UnMaskFork (UMF), which uses Monte Carlo Tree Search to optimize the generation path. The study highlights:
1. Standard AR-based scaling strategies, such as Best-of-N, where temperature is often increased to encourage diversity, is ineffective for MDLMs. Test time scaling of MDLMs needs to derive diversity from structural variations rather than stochastic noise injection.
2. UMF achieves exploration through deterministic action branching. Each action is a choice of a model/temperature/remasking strategy config.
3. UMF switches tokenizers by re-encoding texts.
4. For coding task, reward is defined as the proportion of passed tests. For mathematical tasks, reward is calculated by a PRM.

**Compliance With Llm Reviewing Policy:**

Affirmed.

**Final Justification:**

This paper proposes a novel MCTS-based framework (UMF) for optimizing MDLM generation, with a key innovation in multi-model joint search that expands the boundaries of test-time scaling.

My initial concerns centered on practical feasibility, specifically regarding inference overhead (PRM costs) and compatibility with block diffusion models. However, the authors' rebuttal has effectively addressed these issues. They provided empirical evidence demonstrating that memory and latency overheads are manageable through dynamic weight swapping, and added new experiments confirming compatibility with block diffusion architectures.

Since the methodological novelty is high and the primary weaknesses regarding efficiency and applicability have been convincingly mitigated, I am updating my recommendation to **Weak Accept**.

**Key Questions For Authors:**

1. It is unclear whether this method could be applied to block diffusion models, such as SDAR models. (https://huggingface.co/JetLM/SDAR-8B-Chat)
2. When switching tokenizer, do you maintain a fixed sequence length? How do you manage |MASK| tokens in the process?
3. It is unclear with a single model and different configs (temperature, denoising strategy), how UMF would work with that.
4. Instead of using a PRM model, can internal signals (e.g. confidence) be used as reward?

**Limitations:**

Yes

**Strengths And Weaknesses:**

Strengths:

1. The paper introduces a novel collaborative reasoning paradigm for Masked Diffusion Language Models (MDLMs). It offers a fresh perspective on structured exploration in non-autoregressive generation.

2. The proposed UnMaskFork (UMF) framework is model-agnostic and generalizable. It can be applied to various MDLMs and reasoning tasks (such as coding and mathematics) without being tied to a specific architecture.

3. The submission is well-written, clearly structured, and easy to follow. The motivation behind is well-articulated, and the use of the MCTS framework to formalize the unmasking process is presented clearly.

Weaknesses:

1. A PRM has to be loaded, which introduces extra inference cost and memory footprint.

---

> ### Author Rebuttal · Authors · 2026-03-31
>
> We thank the reviewer for the feedback and for recognizing UMF's novelty and generalizability. Below, we address the raised points and summarize the corresponding updates in our draft.
>
> ---
> ## W1: PRM Overhead and Memory Footprint
> UMF is reward-agnostic and not tied to PRMs. For coding, we use execution-based rewards; we employ a PRM only for math. This is standard in test-time scaling when rule-based verification is unavailable [1,2]; we note this in our current draft.
>
> Regarding inference cost, unmasking forward passes are the main bottleneck; PRM cost is small because it is used for terminal rollouts. We measured the average wall-clock time per problem for the MATH task (Dream-v0 + LLaDA at NFE=3072):
>
> |Total Time(s)|Unmask Time(s)|PRM Time(s)|# PRM Calls|# MDLM Switches|Peak GPU RAM(GiB)|
> |-|-|-|-|-|-|
> |108.93|108.48|0.30|5.00|2.38|46.9|
>
> As shown, unmasking dominates the total time. Importantly, this forward pass is a fixed cost bound to the NFE budget, shared by any test-time scaling method under equivalent constraints.
>
> For memory footprint, model weights can be dynamically offloaded to the CPU. Since Dream-v0, LLaDA, and Qwen2.5-Math-PRM-7B consume ~14.2 GiB, ~14.9 GiB, and ~13.3 GiB of GPU RAM, respectively, in our measurement, keeping only the active model in GPU RAM reduces peak usage to ~15 GiB.
>
> We estimate weight swapping costs using a simple bandwidth bound. The checkpoints are ~15.23 GB (Dream-v0-Instruct-7B) and ~15.30 GB (Qwen2.5-Math-PRM-7B). Assuming an H100 via PCIe Gen5 x16 (64 GB/s) and CPU-resident checkpoints, one sequential swap (evicting Dream, loading Qwen) has a theoretical lower bound of (15.23 + 15.30)/64 ≈ 0.48 seconds.
>
> To complement this bound, we measured actual weight transfer times between pinned CPU memory and the GPU (averaged over 10 trials):
>
> |Model|CPU to GPU (s)|GPU to CPU (s)|
> |-|-|-|
> |Dream-v0|0.56|0.54|
> |LLaDA|0.59|0.57|
> |Qwen2.5-Math-PRM|0.52|0.51|
>
> Although slightly higher than the theoretical bound, a full swap completes within 1.2s. At NFE=3072, a problem requires ~12.38 transfers on average (5×2 for PRM and 2.38 for MDLM switches). This totals ~14.86 seconds per problem, representing an overhead of only ~13.7% of the unmasking time.
>
> Consequently, even in highly memory-constrained environments, multi-model and PRM-guided UMF can be deployed efficiently via fast CPU-GPU weight swapping, avoiding severe latency bottlenecks.
>
> ## Q1: Compatibility with Block Diffusion
> We thank the reviewer for suggesting block diffusion experiments. To demonstrate UMF's compatibility, we evaluated a representative coding block diffusion model, Stable-DiffCoder-8B, combined with LLaDA-8B-Instruct on LiveCodeBench using block diffusion decoding.
>
> To adapt UMF, we decode the sequence block-by-block from left to right. Reaching a predefined unmask ratio triggers an MCTS node generation. Each block is fully unmasked by a single model. We set the block size to 4, matching Stable-DiffCoder-8B's official configuration:
>
> |NFE|768|1536|3072|6144|12288|
> |-|-|-|-|-|-|
> |Pass@1 (%)|19.0|18.0|23.0|28.0|**31.0**|
>
> The single-model baseline Pass@1 at 768 NFE is 19.0%. By applying UMF and scaling compute to 12288 NFE, performance increases to 31.0% (+12 points). We added these details to Appendix A of our current draft.
>
> ## Q2: Sequence Length and Mask Tokens
> We fix the overall decoding budget to 768 tokens. During the tokenizer switch, however, the number of non-special tokens can drift slightly because the two tokenizers segment text differently. We preserve |MASK| token explicitly, and re-tokenize only the non-special text spans.
>
> To assess length variation, we evaluated 100 LiveCodeBench problems at NFE=3072. Switching tokenizers at MCTS nodes introduced an average token length drift of only 1.09%. Empirically, this minor fluctuation does not impact the structural integrity or quality of the generated text.
>
> ## Q3: Single-model Action Variants
> Table 4 provides an ablation study demonstrating that multi-model switching yields the highest performance. When evaluating single-model UMF using varying temperatures or denoising strategies as action types, we found that while single-model configurations successfully outperform standard baselines, multi-model switching remains the best approach among tested action types.
>
> ## Q4: Internal Signals as Reward
> Yes, UMF is highly flexible and supports any scalar reward, including internal signals like model confidence or entropy. We used external verifiers (execution for code [2], PRMs for math [1]) to follow standard test-time scaling protocols for these specific domains.
>
> However, for open-ended generation where external ground-truth is unavailable, employing the MDLM's internal signals as the reward is a viable and natural application of UMF.
>
> [1] Snell et al., 2024, Scaling LLM Test-Time Compute Optimally can be More Effective than Scaling Model Parameters
>
> [2] Brown et al., 2024, Large Language Monkeys: Scaling Inference Compute with Repeated Sampling

---

> > ### Author Rebuttal · Reviewer_pbKi · 2026-04-02
> >
> > Thank you for the comprehensive rebuttal. Most of my concerns have been adequately addressed.
> > Regarding the question about using internal signals as rewards (Q4), although no experiments were provided in this direction, I agree that relying on PRMs and execution-based pass/fail metrics is a more natural and standard approach for code and mathematics scenarios.
> > Given these clarifications, I am updating my overall recommendation score to 4.

---

> > > ### Author Response · Authors · 2026-04-06
> > >
> > > Thank you again for engaging with us during the discussion phase and for your positive update. We truly appreciate the constructive feedback you’ve provided throughout the review process.
> > >
> > > Since some of the points you mentioned overlap with Reviewer Nuc1’s comments, we have expanded on those aspects in our latest response to Reviewer Nuc1. In particular, we added evidence that single-model UMF already outperforms matched baselines, that multi-model UMF improves further beyond independent pairing, and that UMF remains robust under practical verifier mismatch. We hope those additional details will be useful for your final evaluation.

---

### Decision · Program_Chairs · 2026-04-30

**Decision:**

Accept (regular)

**Comment:**

UnMaskFork proposes MCTS-based test-time scaling for masked diffusion LMs via deterministic action branching across multiple MDLM checkpoints, with consistent gains on coding and math benchmarks. Scores are 4/4/3; two reviewers fully resolved after rebuttal added caching ablations, 3-model scaling, wall-clock/memory measurements, and direct comparisons to ReMDM and TReASURe. Reviewer Nuc1's concern about algorithmic versus ensemble gain is legitimate but partially addressed — the rebuttal's evidence that 1-model UMF outperforms ensembled baselines provides meaningful support for the core algorithmic claim.